# A quantitative workflow for modeling diversification in material culture

**Erik Gjesfjeld**[1]*, **Daniele Silvestro**[2], **Jonathan Chang**[3,4], **Bernard Koch**[5], **Jacob G. Foster**[5], **Michael E. Alfaro**[4,6]

**1** McDonald Institute for Archaeological Research, University of Cambridge, Cambridge, England, United Kingdom, **2** Department of Biological and Environmental Sciences, University of Gothenburg, Gothenburg, Sweden, **3** School of Biological Sciences, Monash University, Melbourne, Australia, **4** Department of Ecology and Evolutionary Biology, University of California, Los Angeles, CA, United States of America, **5** Department of Sociology, University of California, Los Angeles, CA, United States of America, **6** Institute for Society and Genetics, University of California, Los Angeles, CA, United States of America

* eg540@cam.ac.uk

**Data Availability Statement:** Data on the production years of automobiles is available at Ebay Parts Compatibility Listings https://pages.ebay.com/motors/compatibility/download.html. A modified version of the Ebay data used in this

## Abstract

Questions about the evolution of material culture are widespread in the humanities and social sciences. Statistical modeling of long-term changes in material culture is less common due to a lack of appropriate frameworks. Our goal is to close this gap and provide robust statistical methods for examining changes in the diversity of material culture. We provide an open-source and quantitative workflow for estimating rates of origination, extinction, and preservation, as well as identifying key shift points in the diversification histories of material culture. We demonstrate our approach using two distinct kinds of data: age ranges for the production of American car models, and radiocarbon dates associated with archaeological cultures of the European Neolithic. Our approach improves on existing frameworks by disentangling the relative contributions of origination and extinction to diversification. Our method also permits rigorous statistical testing of competing hypotheses to explain changes in diversity. Finally, we stress the value of a flexible approach that can be applied to data in various forms; this flexibility allows scholars to explore commonalities between forms of material culture and ask questions about the general properties of cultural change.

## Introduction

One of the most conspicuous products of human cultural diversity is the astonishing number of physical objects created over the course of our history. In this paper, we use the term material culture to broadly describe the totality of physical objects that are created, used and embedded within cultural systems and historical contexts. Given the importance of material 'things' to understanding human history, we aim to demonstrate the potential of a novel quantitative framework for examining long-term changes in material culture. We rely on these methods to address three key questions:

- What is the tempo of change in material culture over time?

analysis is available on FigShare at https://doi.org/
10.6084/m9.figshare.9816326. Data on the
European Neolithic aggregated by EUROEVOL is
available through the University College London
Discovery repository http://discovery.ucl.ac.uk/
1469811/. Additional details about the data can
also be found in Manning et al. or http://doi.org/10.
5334/joad.40. All files and code used in the
analysis can be accessed on Figshare at https://
figshare.com/projects/Quantitative_Workflow_for_
Modeling_Diversification_in_Material_Culture/
68645, including the Supporting Information
available at https://doi.org/10.6084/m9.figshare.
9816731.

**Funding:** EG received support from the Institute for
Society and Genetics at the University of California,
Los Angeles (UCLA), the Renfrew Fellowship from
the McDonald Institute for Archaeological
Research, University of Cambridge and a research
fellowship from Fitzwilliam College. DS received
funding from the Swedish Research Council (2015-
04748) and from the Swedish Foundation for
Strategic Research. JGF, MEA and EG received
funding from a Transdisciplinary Seed Grant
provided by UCLA. EG and MEA received funding
from the Metaknowledge Research Network (ID:
39147) supported by the John Templeton
Foundation.

**Competing interests:** The authors have declared
that no competing interests exist.

- When are the most significant changes in rates of diversification for material culture?

- How do our estimates of diversification compare to existing models of cultural change?

The approach used here builds from a cultural evolutionary perspective to understand long-term changes in material culture. The application of evolutionary principles to understanding material culture has generated substantial debate around diverse topics, such as the role of technology in social change [1, 2], the process of innovation in cultural systems [3–6] and the compatibility of purposeful design or intention with evolutionary principles [7, 8]. One commonality of many existing approaches to material culture is an emphasis on the human subject and their relationship with material things [9]. Within the field of cultural evolution, this 'agent-centered perspective' [10, p.144] often highlights how processes of individual and social learning influence the emergence, diffusion and optimization of innovations. Such studies are clearly important in developing mechanistic models of material culture change, but they are often limited to explaining changes in contemporary objects over short time scales for which detailed information is available. What is currently missing is an approach that is able to utilize the diverse but incomplete archaeological and historical record of material objects or artifacts. As is evident in biology, it is the deep-time record of organisms, identified through fossils, which provides the strongest documentation and evidence for long-term evolutionary change [11].

The fossil record served as the inspiration for seminal studies like *Tempo and Mode in Evolution* [12], which marked a rise in the use of fossils to study macroevolution (the evolution of taxa at or beyond the species level). Macroevolution is now often viewed as the field of evolutionary biology that provides a broad-scale perspective on the forces that shape diversity through time and space. In contrast to microevolutionary studies, macroevolutionary studies generally eschew explicit models of the population dynamics that lead to divergence and speciation in favor of phenomenological models that allow for origination and extinction of species within and across lineages [13, p.227]. Macroevolutionary methods provide a framework for testing whether changes in rates of diversification are explained by discrete or continuous historical factors such as mass extinctions, changing temperatures, or the colonization of new habitats.

The methods we use to answer our research questions offer substantial added value to existing approaches by providing a flexible quantitative framework to examine long-term change using the material record. With our approach, we are able to 1) estimate rates of origination and extinction through continuous time as well as 2) pinpoint historical periods when significant changes occur in material culture diversity. Furthermore, we are able to compare long-term trends in diversity with process models of macro-scale change that derive from evolutionary biology (adaptive radiation, red queen hypothesis [14]), population ecology (predator-prey, Malthusian growth, resilience theory [15]) and economics (dominant designs [16], path dependence [17]). For example, Gjesfjeld et al. [18] find statistical support for the influence of competition and extinction in shaping the diversification history of American car models, similar to expectations from economic models [16]. Overall, the modeling framework presented here provides a unique and flexible set of statistical tools for estimating the tempo of artifact change and exploring the degree to which the diversification dynamics of material culture relate to broader hypotheses of cultural change.

## Units of analysis

The macroevolutionary approach to material culture requires the assumption that "cultural artefacts, like genes or language, reflect their history" [19, p.146]. A similar sentiment is

articulated by Brian Arthur [4]: "What is clear from the historical record is that modern versions of certain technologies do descend from earlier forms"[p.8]. Given the importance of inheritance to material culture, the most appropriate unit of analysis for our study is the *lineage* [10, p.144]. A *lineage* in material culture can be broadly defined as a suite of features that persist in generation after generation of artifacts and display relatively minor changes through time. These features may include technological or morphological traits; lineages can also be defined in terms of economic or social features that demonstrate historical continuity.

We emphasize that our approach does not attempt to create taxonomic units (i.e. lineages) based on shared features. Instead, we use established classification schemes for material culture, which draw on decades of previous work by experts using a variety of evidence. We also do not attempt to infer phylogenetic patterns of inheritance [19–25]; this is not necessary for our scale of analysis, and it can be difficult to infer in many cultural contexts [26]. Broadly speaking, our approach can be used whenever a classification scheme (technological, economic, social) with appropriate lineage-like properties has been created. Furthermore, if multiple taxonomies are available, these results can be compared with each other; performing the same analysis using different classification schemes can help identify robust macroevolutionary patterns and processes.

## Data

We illustrate our approach using two data sets that represent different structures of temporal data. The first is a data set of American car models and their first and last years of production, as identified by their listing in the Ebay Master Vehicle List [27]. We use automobiles as an illustrative example because they represent a highly diverse, long-lived system with extensive and accurate historical data. In total, 3,325 car models from manufacturers with their headquarters located in the United States were used in the analysis, along with their production age ranges (first and last year of production). Car models with gaps in production of over three years were coded as distinct lineages. We refer to this data structure as range data (see Fig 1A), because it only requires an origination date (first year of production) and an extinction date (last year of production), or the present year for car models currently being produced. We focus our analysis at the level of the car model, making the analytic assumption that different car models can be considered as distinct lineages: they have a commercial and cultural identity that persists through time with relatively minor changes in features and physical appearances [18]. This assumption may not be valid for a few car models over the history of American automobiles, but it likely holds for most car models in our analysis.

Our second data set starts with 4,243 radiocarbon dates associated with archaeological pottery produced during the European Neolithic. This data was assembled by the Cultural Evolution of Neolithic Europe project (EUROEVOL) [28] and is openly accessible through the UCL Discovery repository http://discovery.ucl.ac.uk/1469811/. From this larger data set, 22 pottery styles defined by EUROEVOL were selected for analysis based on a minimum sampling criterion along with good geographic and temporal representation as outlined in Manning et al. [29]. In total, 2,626 dates associated with these 22 cultures were chosen for analysis encompassing a range of Neolithic periods (see S1 Table in S1 Data). We refer to the structure of this data as occurrence data: in this data structure, we have a series of time stamps from each culture, but uncertainty about whether the oldest and youngest of our ages represent the "true" times of cultural origination and extinction (see Fig 1B). It is important to note that the EUROEVOL project specifically aimed to compile radiocarbon dates for cultures associated with the Early and Middle European Neolithic; radiocarbon data from later periods (after 4500 cal BP) are more incomplete compared with earlier periods (7500-4500 cal BP). We specifically

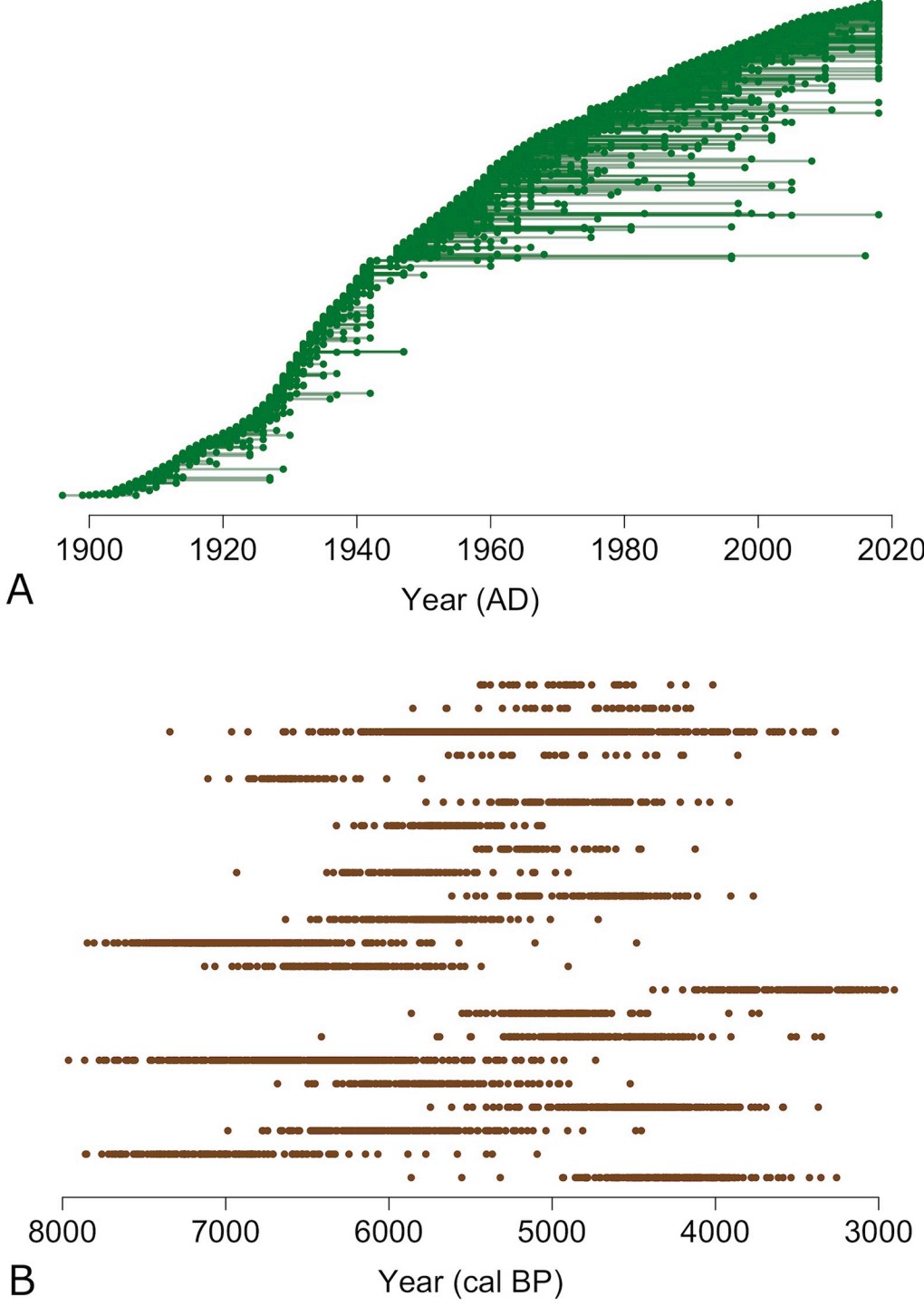

**Fig 1. Visual representation of our two data sets.** A) The American automobiles data is structured as range data with two dates per car model, the first year of production and the last year of production, with consistent production assumed between these years. B) The radiocarbon data from the EUROEVOL project is representative of occurrence data, where each culture has a set of single age estimates. This graphic shows one set of random samples for each of the 22 archaeological cultures as drawn from each sample's calibrated probability distribution. This procedure is performed 100 times to ensure we are capturing the full range of possible ages for each culture.

acknowledge that by using the 22 cultures with the most data, we are limiting our ability to draw interpretations about periods outside of the Neolithic; for example, only 3 of the 2626 dates were associated with the Early Bronze Age period (see S1 Table in S1 Data). Undoubtedly, this transition from the Neolithic to Early Bronze Age is a dynamic period of cultural change involving a wide range of cultural groups from around Europe and the Eurasian steppe. However, the EUROEVOL data set currently lacks enough reliable radiocarbon dates to incorporate this period into our analysis.

Archaeological "cultures" in this data set are based on typological sequences developed by archaeologists from similarities and differences in pottery forms. These traditions of pottery manufacture, such as Linearbandkeramik (LBK) or Corded Ware, represent a classification of the archaeological record into more discrete spatial and temporal units. As highlighted by Manning et al. [29], viewing "cultures" as lineages should not suggest a belief in static and unchanging temporal entities, but rather that the variation between the archaeological cultures being analyzed is greater between them than within them. Therefore, these archaeological "cultures" of pottery traditions may not be useful for investigating local-scale dynamics, but are highly valuable for examining broad scale, macroevolutionary trends [29, p.1066].

All radiocarbon data from the EUROEVOL project were provided as uncalibrated $^{14}$C ages and required calibration prior to analysis. Calibration was performed using the IntCal13 calibration curve [30] as implemented in the `rcarbon` package [31] for the R statistical environment [32]. The process of calibration provides each $^{14}$C sample with a probability distribution of ages based on the variability (or "wiggles") in the calibration curve across the radiocarbon age range. In order to accurately render the probabilistic nature of $^{14}$C data within a time series, each calibrated probability distribution was randomly sampled 100 times and weighted by their calibrated probabilities. In other words, each radiocarbon sample we obtained from the EUROEVOL data set had an additional 100 ages assigned to it based on the probabilities derived through calibration (see Fig 1B and SI for additional details).

By plotting the number of contemporaneous lineages through time, it is clear that our two data sets demonstrate dynamic diversification histories. In both instances, a strong pattern of increasing diversity followed by a sharp decline in diversity can be identified, with the shift for American automobiles occurring in the late 1980s (Fig 2A) and for Neolithic cultures around 5000 years ago (Fig 2B). The longevity of car models and cultures, as measured by their lifespans (Fig 2C and 2D), also demonstrates high variability with most tending towards shorter lifespans; however, a few car models and archaeological cultures do demonstrate much longer lifespans.

## Methodology

### Birth-death process

We modeled diversification based on a birth-death process, and used a Bayesian framework to infer the rates of origination and extinction. The birth-death process is a commonly used model of (macro)evolutionary change, in which origination and extinction are considered as stochastic events occurring at constant or varying rates [33, 34]. Origination and extinction rates can be estimated based on a set of origination and extinction times (i.e. range data) using a probabilistic framework, as shown in Gjesfjeld et al. [18].

If only occurrence data are available, the observed longevity of a lineage, i.e. the time between the oldest and youngest known occurrences of a lineage, is unlikely to capture its entire true duration. This is an inevitable consequence of the fact that occurrences represent an incomplete sample of the lineage history. Thus, inferring origination and extinction rates

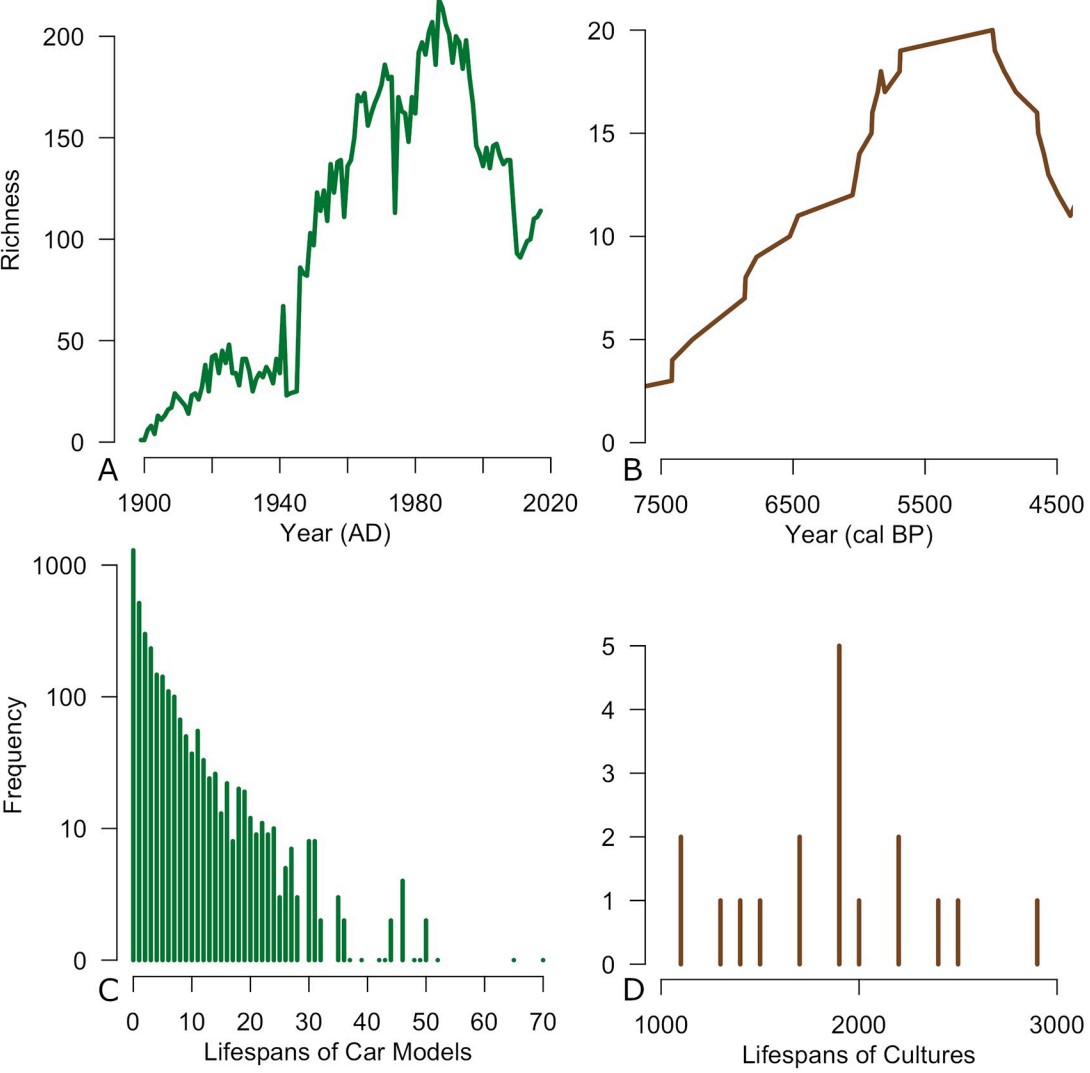

**Fig 2. Change over time in the the richness of American car models (A) and archaeological cultures (B) as well as histograms showing the distribution of lifespans for American car models (C) and archaeological cultures (D).** The car models with the greatest longevity include the Chrysler Town & Country (1946-2016), the Chevrolet Corvette (1953-2018) and the Cadillac DeVille (1953-2005). The archaeological cultures with the longest lifespans are Trichterbecher, Linearbandkeramik and Ertebølle.

from occurrence data requires a joint estimation of the origination and extinction *times*, as outlined in our section on preservation models.

Since a birth-death framework only assumes that the emergence of new lineages is influenced by the number of existing lineages, this method removes strict branching assumptions inherent in alternative evolutionary frameworks (i.e. cladistic or phylogenetic approaches). Our method can therefore be applied to a wider range of data than these earlier frameworks; for example, it can be used with data exhibiting complex inheritance patterns, such as reticulation. A second and equally important contribution is the estimation of both origination and extinction rates. We used a Reversible Jump Markov Chain Monte Carlo (RJMCMC) algorithm [35] to infer origination and extinction rates and their temporal variation. The method starts with the simplest birth-death process of constant origination and extinction rates and

models rate heterogeneity by identifying times of rate shifts. The number and temporal placements of the rate shifts, as well as the origination and extinction rates between shifts, are jointly sampled from their posterior distribution using the RJMCMC algorithm as implemented in the open-source program PyRate [36]. Broadly speaking, the RJMCMC allows us to statistically tease apart stochastic fluctuations of diversity from significant changes in origination and extinction rates. [37]. Strongly supported rate shifts are identified through Bayes factors and are considered more likely to correspond with the occurrence of important historical events or macroevolutionary mechanisms.

The timing of significant rate changes is determined by computing the frequency of sampling a rate shift based on the RJMCMC posterior samples (see [37] for details). An MCMC simulation is performed to evaluate the number of times a rate shift exceeds expectations based on the priors (a uniform distribution on shift times and a Poisson distribution on the number of shifts). Posterior support for rate shifts are evaluated based on standard log Bayes factors thresholds ($2logBF = 2$ indicates positive support, $2logBF = 6$ indicates strong support [38]).

## Preservation models

Because the archaeological and historical record of material culture is almost always incomplete, we model the set of sampled occurrences for each lineage as the result of a Poisson process, in which a preservation rate quantifies the expected number of occurrences per lineage per time unit. We used the PyRate method to couple a Poisson process of preservation with the birth-death model of diversification and jointly estimate 1) the times of origination and extinction for each lineage, 2) the preservation rate, and 3) the origination and extinction rates [33]. The preservation process can be affected by several factors, such as the preservation potential of the material (stone tools versus fiber baskets), sampling effort, geographic range, and age. Thus, preservation rates are likely to change over time and across lineages.

We tested three different models of preservation using a maximum likelihood framework [37]: 1) a homogeneous Poisson Process (HPP) that assumes a constant preservation rate through time; 2) a non-homogeneous Poisson process (NHPP) that assumes preservation rates change along the lifespan of a lineage following a bell-shaped curve (modeled by an extended beta distribution) with rates lower nearer to the times of origination and extinction; 3) a time-variable Poisson process (TPP) in which preservation rates can vary at predefined times of shift. The NHPP model was implemented based on the the expectation that new forms originate from a small initial production, later expanding in their abundance and geographic distribution, and decline toward the end of their lifespan.

All preservation models are additionally able to account for rate heterogeneity among lineages, by assuming that preservation varies among lineages according to an estimated gamma distribution (see [37] for additional details). The relative fit of each model is calculated using the Akaike Information Criterion corrected (AICc) for the number of analyzed lineages. We used PyRate's predefined thresholds to assess the statistical significance of the AICc differences across models.

## Implementation

All birth-death and preservation models used in the analyses are implemented in the open-source program PyRate [36, 37]. This program implements a Bayesian framework to jointly estimate the parameters of the birth-death process (origination and extinction rates through time), and, if necessary, the preservation process (times of origination and extinction and preservation rates). To account for uncertainties in the radiocarbon dating of the occurrences, we

implemented a new function to re-sample the age of each occurrence based on the probability density function obtained from `rcarbon` [31]. PyRate analyses can then be replicated across samples to incorporate age uncertainties in the estimated rates. Tutorials and articles outlining the application of PyRate are currently available [39], with additional resources provided in the supplemental information.

While the estimation of a preservation rate in combination with origination and extinction rates is recommended with occurrence data, this step is unnecessary when the exact times of origination and extinction are known, such as the automobile age range data used here. This greatly decreases the computational burden of the analysis and we therefore implemented a streamlined version of the PyRate method, which is specifically designed for range data. This implementation, named LiteRate, implements the same RJMCMC algorithm available in PyRate to estimate variation in origination and extinction rates through time. If origination and extinction times are known, as is likely for many modern objects, this alternative implementation is able to handle very large data sets (tens of thousands of lineages and more) while reducing the computing time by orders of magnitude compared to the standard PyRate implementation. The program LiteRate is available on GitHub: https://github.com/dsilvestro/LiteRate [40].

## Analysis settings

We analyzed the car data using LiteRate with 20 million RJMCMC iterations, sampling at each 2000th interval (see supplemental information). We analyzed the archaeological data using PyRate. We first tested for the best preservation process, then ran 10 million RJMCMC iterations sampling at every 1000th interval on 100 replicates (see supplemental information). We adjusted the prior settings to accommodate large values of preservation rates based on the maximum likelihood rate estimates. Specifically, the prior on preservation rates was set to a vague exponential prior with a rate parameter of 0.01 to allow for a wide range of preservation rate values, including rates much higher than those expected from fossil data. The list of commands and input data are available as SI. All the parameters of interest, chiefly origination and extinction rates through time, were summarized with a mean and 95% credible intervals.

## Results and discussion

### American automobiles

Results from the analysis of American automobiles indicate that origination and extinction rates closely follow each other through time, a trend also recognized in previous studies using an alternative and more conservative birth-death MCMC (BDMCMC) algorithm [18, 37]. Two dynamic historical periods not previously identified in the more conservative analysis occurred during the early 1920s and the late 1990s-2000s. Starting around 1919, origination and extinction rates both drop substantially for approximately five years before rebounding to previous levels (see Fig 3A). One possible explanation for this is the impact of mass production on the automotive industry, particularly the market dominance of the Ford Model T during this period. A second period of interest not previously identified is the dominant rise of the sport utility vehicle (SUV) during the late 1990s and early 2000s, which is punctuated by the financial crisis of 2007-2008 and subsequent recession. This time period represents one of the few periods where origination and extinction rates do not closely resemble each other. Rather, we see a series of significant shifts in extinction rates but not a significant change in car model origination rates. This result matches the historical evidence; due to the restructuring of General Motors and Chrysler after the financial crisis, many car models (and even some car makes) were discontinued without being replaced, as production shifted to focus on high-

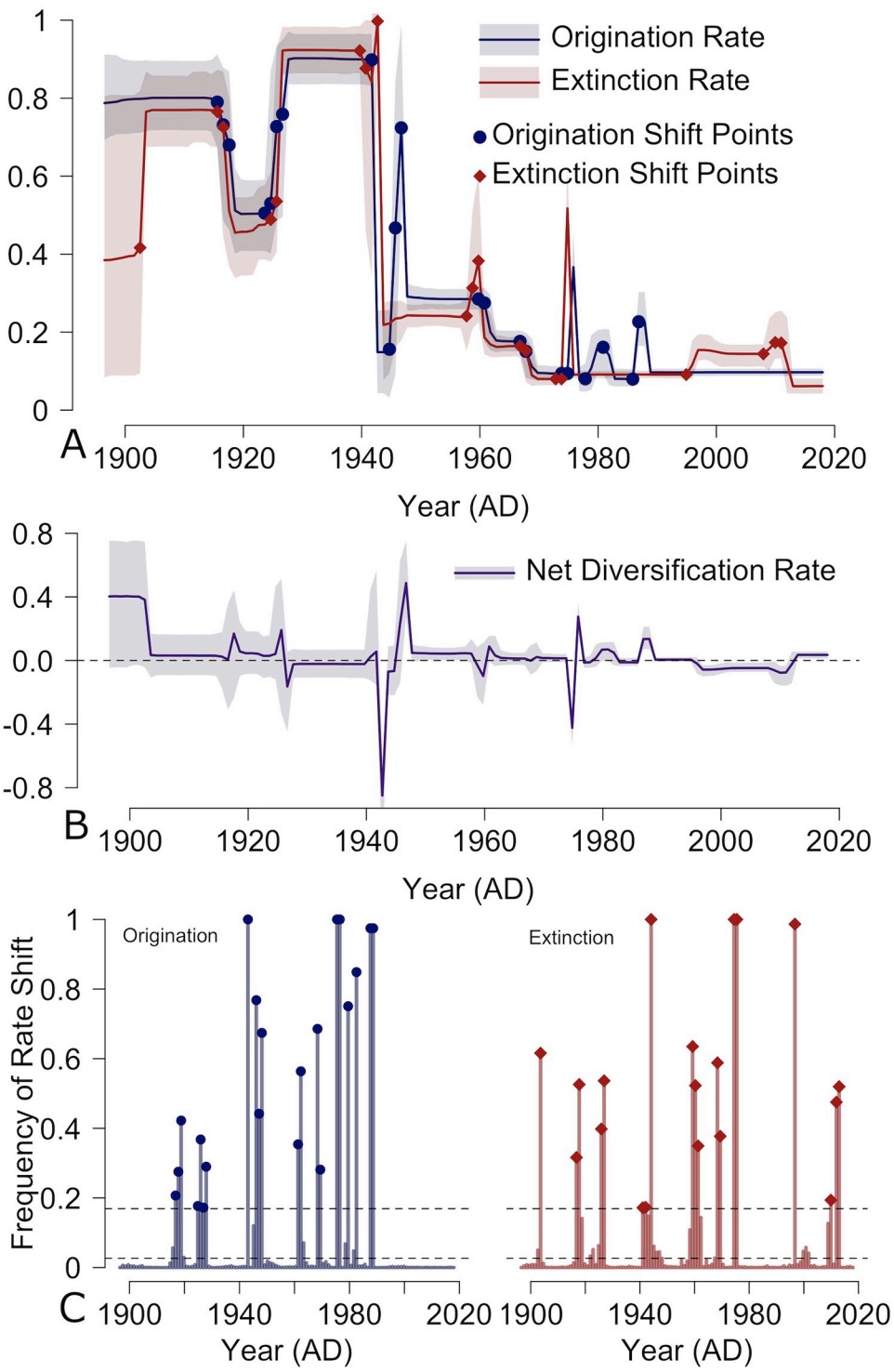

**Fig 3. Rates of diversification through time for American car models.** A) Solid lines indicate mean origination and extinction rate for each year with shaded areas representing 95% highest probability densities. B) Changes in the net diversification rate through time (origination rate—extinction rate) with the dotted line indicated net diversification of 0. C) Frequency of rate shifts based on posterior samples for each time unit of our analysis. Dotted lines indicate the strength of evidence for the rate shift frequency as corresponding to the standard natural log Bayes Factors thresholds [38] of 2 (lower dotted line) and 6 (higher dotted line).

profit truck and SUV models. Ultimately, we can infer that the diversification history of American car models is influenced by a number of factors. Some changes, such as the rise of the Model T or SUVs, are likely associated with changing consumer preferences or fashion cycles. However, we also see major fluctuations in car model diversity that coincide with (and are likely driver by) external changes in the socio-economic environment, such as the declines in car model diversity associated with World War II and the 1970s Arab Oil Crises.

A second outcome of our updated analysis using RJMCMC is that the negative net diversification rate between the mid-1990s to the mid-2010s has rebounded over the last few years to be a small, but positive net diversification rate. This suggests that for the first time since the mid-1990s, the total diversity of American car models is increasing as more new car models are originating than are being discontinued. The most likely cause for this gain in diversity is the emergence of a slow, but growing electric car market [41], a trend predicted in a previous publication [18].

Overall, the results of our updated analysis are encouraging. We increased temporal resolution but also replicated the correlation between origination and extinction rates. As with the earlier analysis [18], our results are consistent with historical research that highlights the dynamism of the American automobile industry and the importance of both external and internal factors in driving diversification dynamics.

## Neolithic Europe

Results of the preservation model test indicate the best fitting preservation model for our Neolithic Europe occurrence data is a non-Homogeneous Poisson Process (NHPP) (see Table 1). As discussed above, this model assumes that preservation rates change along the lifespan of a lineage following a bell-shaped distribution. A gamma model was also implemented to account for rate heterogeneity across lineages.

Estimates of origination and extinction rates of European Neolithic cultures demonstrate three distinct phases (Fig 4A). The first phase (7500-5800 cal BP) is one of substantial growth in the richness of cultures with a high origination rate and low extinction rate that suggests rapid linear growth in the number of archaeological cultures / traditions. The second phase (5800-5000) is marked by a major transition with a dramatic slowdown in the origination rate leading to a period of limited growth in the number of archaeological cultures. This period witnesses the "tipping point" where the diversity of pottery styles slows and begins to decline, approximately around 5500 cal BP. The final phase (5000-4500) highlights a dramatic and sustained rise in the extinction rate with low rates of origination. The cumulative outcome of these phases is a gradual decline in net diversification starting approximately 5800 cal BP and continuing to the end of the Neolithic period (Fig 4B).

The value of our approach is the ability to disentangle the diversity of pottery types into its constituent parts of origination and extinction. As shown in Fig 5, cultural diversity, as measured by the richness of pottery forms, steadily increases from 7500 cal BP to 5800 cal BP, plateaus between 5800 to 5000 cal BP, and then declines after 5000 (Fig 5A). However, this diversity trajectory could be the product of different origination and extinction relationships,

**Table 1. Results of preservation model test indicating significant support for the NHPP preservation model.** TPP model used a three-period model broadly based on European archaeological phases (see supplemental information for additional details).

| Model | Maximum Likelihood | Preservation Rate(s) | AICc | dAICc |
|---|---|---|---|---|
| NHPP | -3931.17 | 79.3 | 7864 | - |
| TPP | -5180.35 | 105.05,101.48,75.30 | 10368 | 2504*** |
| HPP | -5235.99 | 87.53 | 10474 | 2610*** |

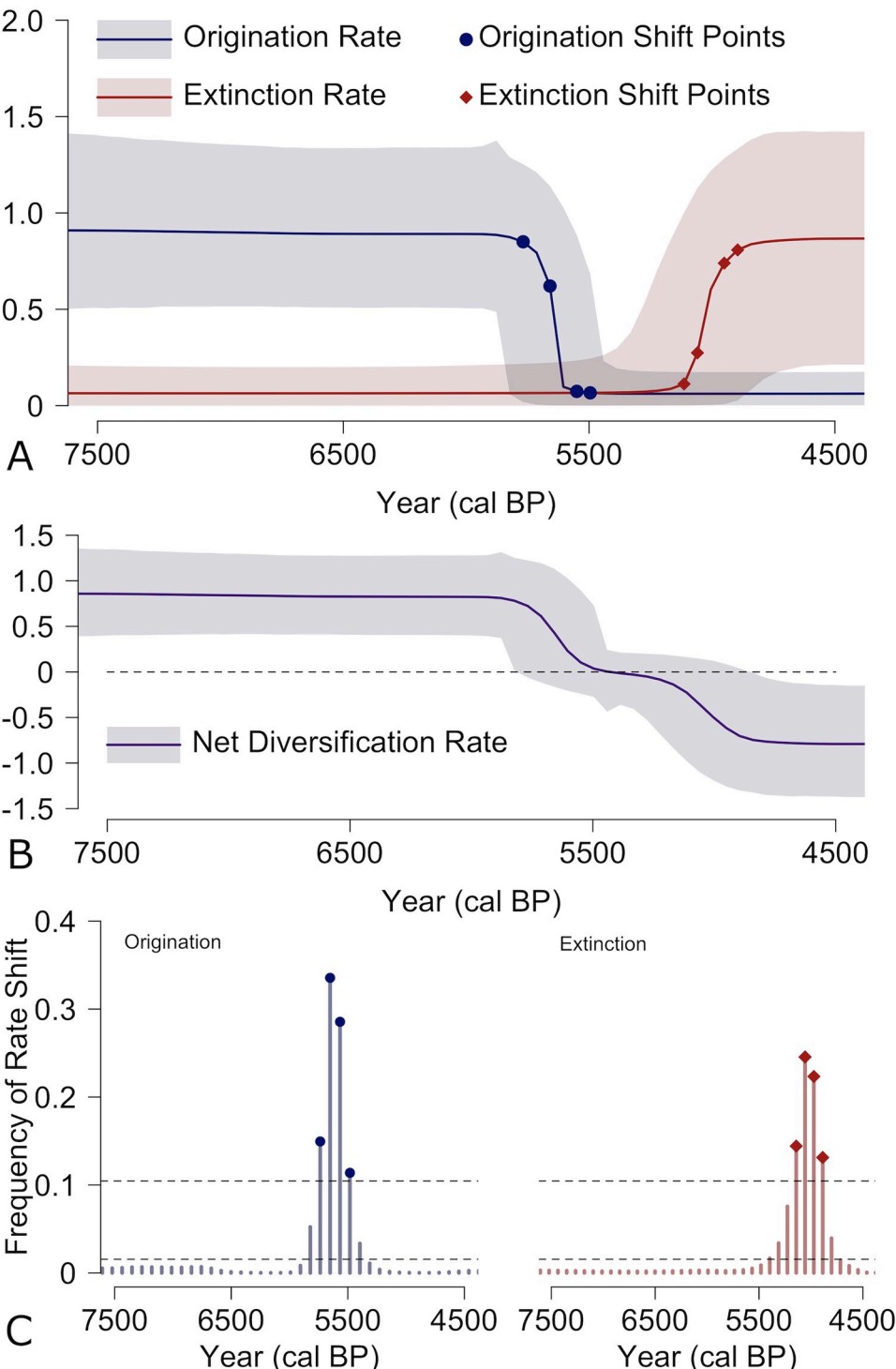

**Fig 4. Rates of diversification through time for cultures of the European Neolithic.** A) Solid lines indicate mean origination and extinction rate for each year with shaded areas representing 95% highest probability densities. B) Changes in the net diversification rate through time (origination rate—extinction rate) with the dotted line indicated net diversification of 0. C) Frequency of rate shifts based on posterior samples for each time unit of our analysis. Dotted lines indicate the significance of the rate shift frequency as corresponding to the logged Bayes Factors of 2 (lower dotted line) and 6 (higher dotted line).

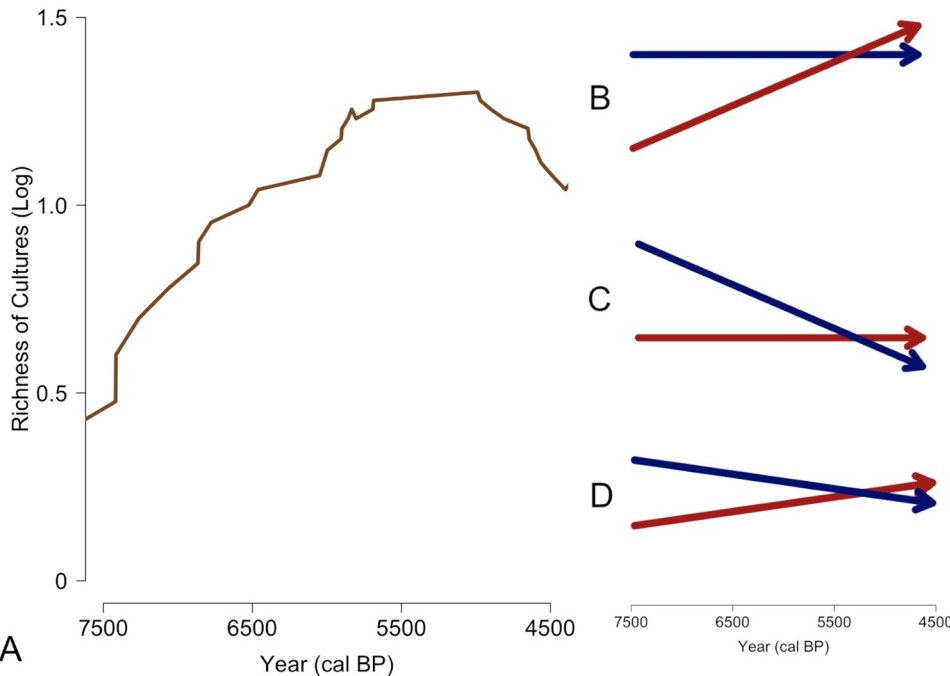

**Fig 5. Overall log richness of archaeological cultures (i.e. pottery types) (A) with three scenarios that would produce a similar pattern of rising diversity followed by a decline in diversity.** This includes a A) a constant origination rate (blue) and increasing extinction rate (red); B) a declining origination rate and constant extinction rate; and C) a declining origination rate and increasing extinction rate.

such as a stable origination rate and increasing extinction rate (Fig 5B), a stable extinction rate and a decreasing origination rate (Fig 5C), or a combination of a declining origination rate and a increasing extinction rate (Fig 5D).

The pattern estimated using our approach (Fig 4A) suggests the most likely diversification scenario: the combination of a declining origination rate and increasing extinction rate. This is a valuable insight, as it mirrors diversification dynamics associated with the red queen hypothesis [42]. Broadly speaking, the red queen hypothesis [14] suggests that in a changing or deteriorating environment, simply maintaining existing diversity requires constant adaptation to environmental conditions. Red queen evolutionary dynamics can drive diversity loss if origination rates decrease and do not compensate for increasing extinction [42]. Empirical fossil data from terrestrial mammals highlight how both origination and extinction rates impact the diversity trajectory, showing that decreasing origination rates are just as important as increasing extinction rates in determining the rise and demise of biological clades [42, 43].

The underlying processes that influence origination and extinction rates are undoubtedly a combination of both internal and external factors. In our archaeological example, the precise factors that produce the red queen pattern can be difficult to ascertain, but we can broadly hypothesize a deterioration in the socio-ecological environment. Recently, Colledge et al. [44] provide evidence for this deterioration by demonstrating a decline in agricultural production due to a decrease in the soil fertility of arable lands at approximately 5500 cal BP. These changes in agriculture also correlate with declines in population densities highlighted in summed radiocarbon distribution models [45] as well as our temporal estimates of shifts in material culture origination rates (Fig 4C). These pieces of evidence suggest a combination of internal (population growth) and external (soil fertility) factors coming together between

5800-5500 cal BP to create a "tipping point" that disrupts the existing cultural system and drives the loss of material culture diversity.

## Conclusion

The fossil record has proven to be a valuable source of evidence for biological evolution. We view the archaeological and historical record of material culture as equally valuable for understanding cultural evolution. Here, we presented a novel workflow that models diversification dynamics using temporal data from different kinds of material culture. We consider this quantitative framework as providing substantial added value in examining material culture change based on four features. The first is that our approach builds from a flexible evolutionary assumption that the current diversity of material culture is a product of previous diversity. This is different from the application of a cladistic or phylogenetic approach, which like our approach assumes inheritance, but often includes a more explicit assumption about the underlying bifurcating process of origination, which has been criticized [19].

The second feature of our approach is its implementation in a hierarchical and fully probabilistic framework that provides robust statistical evidence for testing macroevolutionary models and for quantifying the uncertainty around diversification rate estimates. The Bayesian RJMCMC algorithm implemented here allows us to explore a potentially infinite number of models, while distinguishing between noise and signal in temporal changes of birth and death rates. This allows for models of change that account for both diversification and preservation or sampling processes. In addition, it permits the use of incomplete archaeological and historical data by explicitly modeling the heterogeneity of sampling for each lineage.

Third, in this framework we model the dynamics of origination, but also the equally dynamic, far less well-understood patterns of extinction. Just like the majority of biological organisms, most innovations in human history are functionally extinct. The process of extinction or loss is often assumed to be part of an adaptive process in which a new and potentially better lineage emerges that out-competes the previous ones. Our approach provides a framework to explore whether macro-scale models used to explain diversity loss from evolutionary biology [14, 42, 46] are also applicable to understanding cultural extinction [47].

Finally, our approach makes it tractable to leverage the vast, but incomplete archaeological and historical information about changes in material culture through time. With the exploding availability of digital databases of artifacts, often containing associated temporal data, we see our methodology using age ranges and occurrences as flexible enough to be applied to a wide range of potential data sets, yet robust enough to easily compare between different forms. This creates opportunities to evaluate a wide range of material culture forms and explore potential commonalities in the mode and tempo of change. Overall, we argue that by using robust quantitative methods for modeling the macroevolutionary dynamics of material culture change, we can complement existing cultural evolutionary studies by providing robust historical evidence for the patterns and processes of cultural change.

## Supporting information

**S1 Data.**
(PDF)

## Author Contributions

**Conceptualization:** Erik Gjesfjeld, Daniele Silvestro, Jonathan Chang, Bernard Koch, Jacob G. Foster, Michael E. Alfaro.

**Data curation:** Erik Gjesfjeld.

**Formal analysis:** Erik Gjesfjeld.

**Funding acquisition:** Daniele Silvestro, Michael E. Alfaro.

**Investigation:** Erik Gjesfjeld, Daniele Silvestro, Jonathan Chang.

**Methodology:** Erik Gjesfjeld, Daniele Silvestro, Bernard Koch, Michael E. Alfaro.

**Supervision:** Erik Gjesfjeld, Michael E. Alfaro.

**Writing – original draft:** Erik Gjesfjeld, Daniele Silvestro.

**Writing – review & editing:** Erik Gjesfjeld, Daniele Silvestro, Jonathan Chang, Bernard Koch, Jacob G. Foster, Michael E. Alfaro.

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
