## [Decision Letter · Decision Letter 0]

14 Oct 2019

PONE-D-19-25807

A quantitative workflow for modeling diversification in technological systems

PLOS ONE

Dear Dr. Gjesfjeld,

Thank you for submitting your manuscript to PLOS ONE. After careful consideration, we feel that it has merit but does not fully meet PLOS ONE’s publication criteria as it currently stands. Therefore, we invite you to submit a revised version of the manuscript that addresses the points raised during the review process.

We would appreciate receiving your revised manuscript by Nov 28 2019 11:59PM. To enhance the reproducibility of your results, we recommend that if applicable you deposit your laboratory protocols in protocols.io, where a protocol can be assigned its own identifier (DOI) such that it can be cited independently in the future. For instructions see: http://journals.plos.org/plosone/s/submission-guidelines#loc-laboratory-protocols

We look forward to receiving your revised manuscript.

Kind regards,

Peter F. Biehl, PhD

Academic Editor

PLOS ONE

**Journal Requirements:**

**Additional Editor Comments (if provided):**

Your manuscript has now been seen by two referees, whose comments are appended below. You will see from these comments that while the referees find your work of potential interest, they have raised substantial concerns that must be addressed. In light of these comments, we cannot accept the manuscript for publication, but would be interested in considering a revised version that addresses these serious concerns.

We hope you will find the referees' comments useful as you decide how to proceed. Should presentation of further data and analysis allow you to address these criticisms, we would be happy to look at a substantially revised manuscript. However, please bear in mind that we will be reluctant to approach the referees again in the absence of major revisions.

**Comments to the Author**

1. Is the manuscript technically sound, and do the data support the conclusions?

Reviewer #1: Yes

Reviewer #2: Yes

2. Has the statistical analysis been performed appropriately and rigorously? 

Reviewer #1: Yes

Reviewer #2: Yes

3. Have the authors made all data underlying the findings in their manuscript fully available?

Reviewer #1: Yes

Reviewer #2: Yes

4. Is the manuscript presented in an intelligible fashion and written in standard English?

Reviewer #1: Yes

Reviewer #2: Yes

5. Review Comments to the Author

Reviewer #1: The authors present an intriguing method for examining diachronic change and diversity in technological systems. The article is well-written and easy to follow. The methods are discussed very clearly in the article and accompanying supplement, enabling readers to replicate their methodology.

I have concerns with the choice of archaeological case study, which relies on general Neolithic cultural typology. This somewhat undermines the proposed focus on particular technological assemblages. The modern case study and discussion involving automobiles is far more instructive, showing the model's ability to illustrate the emergence and "extinction" of automotive models and types within a broader socioeconomic context. However I understand that this is an initial explication of the method and hope more detailed archaeological applications will follow.

The choice to take the discussion into a demographic direction is a mistake, in my opinion. Demographic models based on radiocarbon data are poor in terms of their resolution, theoretically contentious in terms of their founding assumptions, and often correlate poorly with archaeological and environmental proxies in those regions where different classes of data can be compared. I think that, at the very least, the authors should acknowledge these limitations and address some of the following points listed below.

Line 127: change "investigate" to "investigating"

Line 129: "data" is a plural noun; change "was" to "were"

Lines 134-139: From a radiocarbon perspective I found this explanation to initially be a little confusing. As I read on I got the gist of it - that you are resampling each probability distribution so that it can be approximated by discrete data points. It would help to just clarify its introduction a little more; something along the lines of "In order to accurately render the probabilistic nature of 14C data within a time series, each calibrated probability distribution was randomly sampled 100 times [...]." I appreciate your method... it is far better than dealing with means or medians as many studies do.

Line 144: The Early Bronze Age transition is an important point of discussion that you seem to ignore in this paper (more below).

Lines 321-322: Why? Can you demonstrate that the number or "richness" of cultural groups is correlated with population? Your demographic proxy is based on radiocarbon, which is at best a crude proxy for population and is subject to extensive sampling bias. "Boom and bust" is in vogue in many macro-scale analyses performed by Shennan and his colleagues, but where has it been verified in regional archaeological data sets? This deserves expansion and discussion, because use of 14C in this manner is by no means universally accepted among archaeologists or radiocarbon specialists (cf. Attenbrow and Hiscock 2015; Contreras and Meadows 2014; Torfing 2016; and, in North America, the whole saga of comments and responses associated with Buchanan et al. 2008).

Line 327: Can you better explain how Puleston et al.'s research is applicable to Neolithic Europe? It is an apt conceptual model, but it is assuming hard limits to growth based on available resources and territory in a static, closed system. Settlement patterns in Neolithic Europe tend towards constant movement and expansion into peripheral regions, behavior which continued well into the Eneolithic. It is perhaps applicable during the 4th millennium BC, when widespread changes to more mobile subsistence regimes occur in the run-up to the EBA transition and a case could be made that potential natural increase is depressed. Without further explanation it kind of seems like you're arbitrarily imposing a model here, trying to shove a square peg in a round hole.

Lines 330-334: Is the "population" being discussed describing the statistical population of archaeological cultures, or the inferred actual human population? I don't disagree with these trends on a very general level (then again, at these levels, many phenomena will take on a vaguely Gaussian distribution), but you're also only telling part of the story. At 5000 cal BP, where you are showing (correctly) a precipitous drop in Neolithic cultural groups, there is a rapid influx of Early Bronze Age groups coming from the Eurasian steppe. This at least bears mentioning: that these groups during the Terminal Neo-Eneolithic are not existing in a closed system.

Reviewer #2: This is an excellent paper that makes a novel contribution to empirical studies of cultural evolution and in doing so contributes new insights. I do though have one significant substantive concern: it's not clear to that the diversification processes concerned relate to technology. In the case of the American automobiles isn't most of it just fashion change? In the case of the Neolithic cultures there's little evidence that the diversification relates to subsistence, the most important technology. It is basically a pattern of cultural descent and differentiation arising from spatial expansion and growing isolation by distance of the groups concerned.

If the authors want to insist that their paper is about diversification in technology rather than fashion then they need to show how this is the case

6. PLOS authors have the option to publish the peer review history of their article (what does this mean?). If published, this will include your full peer review and any attached files.

Reviewer #1: No

Reviewer #2: No

---

## [Author Response · Author response to Decision Letter 0]

10 Dec 2019

Response to Reviewer Comments: A quantitative workflow for modeling diversification in material culture (Gjesfjeld, Silvestro, Chang, Koch, Foster and Alfaro)

We would first like to say thank you to the reviewers for very helpful and insightful comments that we all believe strengthened the manuscript. Based on the reviewer suggestions, a number of significant changes were made to the manuscript and the details of these changes can be found in the italicized responses below. 

Reviewer #1: The authors present an intriguing method for examining diachronic change and diversity in technological systems. The article is well-written and easy to follow. The methods are discussed very clearly in the article and accompanying supplement, enabling readers to replicate their methodology.

I have concerns with the choice of archaeological case study, which relies on general Neolithic cultural typology. This somewhat undermines the proposed focus on particular technological assemblages. The modern case study and discussion involving automobiles is far more instructive, showing the model's ability to illustrate the emergence and "extinction" of automotive models and types within a broader socioeconomic context. However I understand that this is an initial explication of the method and hope more detailed archaeological applications will follow.

The choice to take the discussion into a demographic direction is a mistake, in my opinion. Demographic models based on radiocarbon data are poor in terms of their resolution, theoretically contentious in terms of their founding assumptions, and often correlate poorly with archaeological and environmental proxies in those regions where different classes of data can be compared. I think that, at the very least, the authors should acknowledge these limitations and address some of the following points listed below.

Response: See Below

Line 127: change "investigate" to "investigating"

Response: Fixed

Line 129: "data" is a plural noun; change "was" to "were"

Response: Fixed

Lines 134-139: From a radiocarbon perspective I found this explanation to initially be a little confusing. As I read on I got the gist of it - that you are resampling each probability distribution so that it can be approximated by discrete data points. It would help to just clarify its introduction a little more; something along the lines of "In order to accurately render the probabilistic nature of 14C data within a time series, each calibrated probability distribution was randomly sampled 100 times [...]." I appreciate your method... it is far better than dealing with means or medians as many studies do.

Response: Yes, this is exactly what we are doing and the text has incorporated the suggested text above

Line 144: The Early Bronze Age transition is an important point of discussion that you seem to ignore in this paper (more below).

-Yes, we agree but this is largely because the EUROEVOL data set does not have reliable data on the Early Bronze Age. In speaking with the PI of the EUROEVOL project (Stephen Shennan), we were strongly advised to not use radiocarbon dates in the EUROEVOL data set associated with the Early Bronze Age as this was not the focus of their inquiry and there was no systematic attempt to acquire dates that are associated with this period, unlike the Neolithic samples. We are by no means discounting the importance of the Neolithic-Early Bronze Age transition but simply did feel that we had the necessary data to analyze this period, both in terms of radiocarbon dates and their associated pottery types. Furthermore, we feel the main contribution of the paper is a methodological one and not about the complexities of the Eneolithic / Chacolithic period (which none of us are qualified to discuss in great detail). 

Lines 321-322: Why? Can you demonstrate that the number or "richness" of cultural groups is correlated with population? Your demographic proxy is based on radiocarbon, which is at best a crude proxy for population and is subject to extensive sampling bias. "Boom and bust" is in vogue in many macro-scale analyses performed by Shennan and his colleagues, but where has it been verified in regional archaeological data sets? This deserves expansion and discussion, because use of 14C in this manner is by no means universally accepted among archaeologists or radiocarbon specialists (cf. Attenbrow and Hiscock 2015; Contreras and Meadows 2014; Torfing 2016; and, in North America, the whole saga of comments and responses associated with Buchanan et al. 2008).

Line 327: Can you better explain how Puleston et al.'s research is applicable to Neolithic Europe? It is an apt conceptual model, but it is assuming hard limits to growth based on available resources and territory in a static, closed system. Settlement patterns in Neolithic Europe tend towards constant movement and expansion into peripheral regions, behavior which continued well into the Eneolithic. It is perhaps applicable during the 4th millennium BC, when widespread changes to more mobile subsistence regimes occur in the run-up to the EBA transition and a case could be made that potential natural increase is depressed. Without further explanation it kind of seems like you're arbitrarily imposing a model here, trying to shove a square peg in a round hole.

Lines 330-334: Is the "population" being discussed describing the statistical population of archaeological cultures, or the inferred actual human population? I don't disagree with these trends on a very general level (then again, at these levels, many phenomena will take on a vaguely Gaussian distribution), but you're also only telling part of the story. At 5000 cal BP, where you are showing (correctly) a precipitous drop in Neolithic cultural groups, there is a rapid influx of Early Bronze Age groups coming from the Eurasian steppe. This at least bears mentioning: that these groups during the Terminal Neo-Eneolithic are not existing in a closed system.

Response: Once again, we agree with the reviewer’s concern about our interpretation of the origination and extinction rates in a demographic framework. After further consideration, we have removed this paragraph and our discussion of Puleston’s invisible cliff model. We agree that the conceptual and methodological leap between the two is a bit tenuous and admittedly a little forced (as noted by the reviewer). 

We have decided to take our discussion in a different direction and interpret our diversification patterns within existing macroevolutionary models, as we believe this is a more justifiable and clearer connection given our methodological approach. Specifically, we now highlight how diversity loss (or extinction) appears to be driven by the combination of declining origination rates and increasing extinction rates. This pattern is very similar to expectations of a “red queen” extinction scenario, as suggested by Quental and Marshall in their 2013 Science paper. We see this interpretation as one is more directly connected to our results and does not rely on demographic estimates from radiocarbon data, which admittedly can be controversial, as suggested by the reviewer. We do draw parallels between our red queen interpretation and recent archaeological evidence that suggests a deteriorating agricultural environment, perhaps caused by substanial population growth (Colledge et al. 2019). That being said, we did not want to produce a lengthy discussion on the European Neolithic, as this is not the goal of this paper and we are not qualified to discuss this in detail. We simply want to show that our method can produce new insights into material culture diversity patterns that can potentially help to further inform the evolutionary and historical processes underlying these patterns. 

Reviewer #2 

This is an excellent paper that makes a novel contribution to empirical studies of cultural evolution and in doing so contributes new insights. I do though have one significant substantive concern: it's not clear to that the diversification processes concerned relate to technology. In the case of the American automobiles isn't most of it just fashion change? In the case of the Neolithic cultures there's little evidence that the diversification relates to subsistence, the most important technology. It is basically a pattern of cultural descent and differentiation arising from spatial expansion and growing isolation by distance of the groups concerned. If the authors want to insist that their paper is about diversification in technology rather than fashion then they need to show how this is the case.

Response: The reviewer brings up an excellent point and one that admittedly we struggled with when writing this paper. We often debated whether "technology" was the correct term for what we were investigating or whether we should use a more generic term like “cultural things”, which would more broadly include fashion change, or more neutral processes of change. In this revision, we opted to use the term “material culture” rather than “technologies”, as the latter may be too restrictive of term and comes with many additional connotations and variable definitions (although we see technologies as a subset of material culture). We hope that the term "material culture" reflects a broader conceptualization of physical objects and better signals that we are open to a range of processes that can produce diversification patterns that our methods identify, including fashion change. 

As for the comment about fashion change in American automobiles, I would argue that if fashion change was the underlying driver of diversity in American car models, we would expect a fairly constant rate of turnover and therefore diversification rates, which is the initial assumption of our modeling framework. We are assuming (perhaps wrongly) that neutral fashion cycles would produce constant or linear origination and extinction rates through time, as the turnover rate in fashion cycles tends to be fairly linear. However, in both our examples we find our best fitting model is not one of constant change, but rather models that have major shifts in diversification rates during certain historical times. We would further argue that fashion change (in a neutral sense) is likely influencing changes in diversity, but these cycles are often disrupted by "external" changes in the social and economic environment. In the cars example, these disruptions seem to occur relatively frequently, whereas in the Neolithic cultures, there may be long period (7500-5800 cal BP), where the relationship between origination rates and extinction rates is fairly constant. That being said, fashion cycles can be difficult to measure using artifact richness, especially without reliable frequency information. We have added a few sentences (Lines 281-287) that highlight the above point, that the diversification pattern of automobiles that we see is most likely a combination of changing consumer preferences that are punctuated by major “external” events (WWII, Arab Oil Crises). 

In response to comments from the first reviewer, we have also adjusted our discussion about comparing diversification rates with demographic processes. In the revised version, we suggest that our diversification pattern is most similar to a “red queen” scenario as demonstrated by Quental and Marshall (2013). The process underlying a red queen pattern is often considered to be environmental deterioration and a failure for organisms to adapt to changing conditions. In this respect, changes in pottery diversity (and population) do correlate to changes in agricultural productivity and subsistence practices as highlighted in the recent article by Colledge et al. 2019. I would agree with the reviewer that one likely underlying process in the first phase (7500-5800 cal BP) is spatial expansion and then growing isolation by distance, which would produce steady gains in diversity. The insights produced from our approach is that this process has a fairly quick and dramatic end, potentially due to worsening social and ecological conditions, as argued above. Broadly speaking, our approach is aimed at quantifying patterns of diversification, which do not directly identify specific cultural or demographic processes but rather aim to narrow the range of potential processes that might explain the pattern.

---

## [Decision Letter · Decision Letter 1]

23 Dec 2019

A quantitative workflow for modeling diversification in material culture

PONE-D-19-25807R1

Dear Dr. Gjesfjeld,

We are pleased to inform you that your manuscript has been judged scientifically suitable for publication and will be formally accepted for publication once it complies with all outstanding technical requirements.

With kind regards,

Peter F. Biehl, PhD

Academic Editor

PLOS ONE

Additional Editor Comments (optional):

Reviewers' comments:

Reviewer's Responses to Questions

**Comments to the Author**

1. If the authors have adequately addressed your comments raised in a previous round of review and you feel that this manuscript is now acceptable for publication, you may indicate that here to bypass the “Comments to the Author” section, enter your conflict of interest statement in the “Confidential to Editor” section, and submit your "Accept" recommendation.

Reviewer #1: All comments have been addressed

Reviewer #2: All comments have been addressed

2. Is the manuscript technically sound, and do the data support the conclusions?

Reviewer #1: Yes

Reviewer #2: Yes

3. Has the statistical analysis been performed appropriately and rigorously? 

Reviewer #1: Yes

Reviewer #2: Yes

4. Have the authors made all data underlying the findings in their manuscript fully available?

Reviewer #1: Yes

Reviewer #2: Yes

5. Is the manuscript presented in an intelligible fashion and written in standard English?

Reviewer #1: Yes

Reviewer #2: Yes

6. Review Comments to the Author

Reviewer #1: (No Response)

Reviewer #2: (No Response)

7. PLOS authors have the option to publish the peer review history of their article (what does this mean?). If published, this will include your full peer review and any attached files.

Reviewer #1: No

Reviewer #2: No

---

## [Editor Report · Acceptance letter]

6 Jan 2020

PONE-D-19-25807R1 

A quantitative workflow for modeling diversification in material culture 

Dear Dr. Gjesfjeld:

I am pleased to inform you that your manuscript has been deemed suitable for publication in PLOS ONE. Congratulations! Your manuscript is now with our production department. 

With kind regards,

on behalf of

Dr. Peter F. Biehl 

Academic Editor

PLOS ONE